# Current Trends and Beyond Conventional Approaches: Advancements in Breast Cancer Surgery through Three-Dimensional Imaging, Virtual Reality, Augmented Reality, and the Emerging Metaverse

**DOI:** 10.3390/jcm13030915

**Published:** 2024-02-05

**Authors:** Weronika Magdalena Żydowicz, Jaroslaw Skokowski, Luigi Marano, Karol Polom

**Affiliations:** 1Department of General Surgery and Surgical Oncology, “Saint Wojciech” Hospital, “Nicolaus Copernicus” Health Center, Jana Pawła II 50, 80-462 Gdańsk, Poland; veronikazydowicz@wp.pl (W.M.Ż.); j.skokowski@amisns.edu.pl (J.S.); 2Department of Medicine, Academy of Applied Medical and Social Sciences, Akademia Medycznych I Spolecznych Nauk Stosowanych (AMiSNS), 2 Lotnicza Street, 82-300 Elbląg, Poland; k.polom@amisns.edu.pl; 3Department of Gastrointestinal Surgical Oncology, Greater Poland Cancer Centre, Garbary 15, 61-866 Poznan, Poland

**Keywords:** augmented reality, virtual reality, 3D, breast cancer, breast operations, image-guided surgery, cancer imaging

## Abstract

Breast cancer stands as the most prevalent cancer globally, necessitating comprehensive care. A multidisciplinary approach proves crucial for precise diagnosis and treatment, ultimately leading to effective disease management. While surgical interventions continue to evolve and remain integral for curative treatment, imaging assumes a fundamental role in breast cancer detection. Advanced imaging techniques not only facilitate improved diagnosis but also contribute significantly to the overall enhancement of breast cancer management. This review article aims to provide an overview of innovative technologies such as virtual reality, augmented reality, and three-dimensional imaging, utilized in the medical field to elevate the diagnosis and treatment of breast cancer. Additionally, the article delves into an emerging technology known as the metaverse, still under development. Through the analysis of impactful research and comparison of their findings, this study offers valuable insights into the advantages of each innovative technique. The goal is to provide physicians, surgeons, and radiologists with information on how to enhance breast cancer management.

## 1. Introduction

Breast cancer remains the most frequently diagnosed cancer and a leading cause of female mortality [1,2,3]. Late-stage detection persists despite preventive measures like screening mammography, resulting in disproportionately high mortality rates. In 2020, WHO reported 2.3 million new cases, with about 30% succumbing to the disease [4]. Over the past decade, progress has aimed to improve the patient’s quality of life and survival rates [5,6], emphasizing the pivotal role of early detection [7]. In this progress landscape, advanced imaging modalities—virtual reality (VR), augmented reality (AR), and three-dimensional imaging (3D)—have emerged, refining breast cancer treatment through precise tumor diagnosis and improved surgical interventions. These innovations reshape healthcare capabilities on a broader scale [8,9,10]. Beyond the clinic, these technologies extend into education, offering possibilities for online training [11,12] and interactive learning experiences [9,13]. They transcend treatment confines, influencing patient rehabilitation and advancing comprehensive healthcare [10]. This review explores diagnostic techniques, focusing on VR, AR, and 3D imaging in breast cancer management, aiming to provide nuanced insights into transformative healthcare experiences.

## 2. Materials and Methods

A review article serves as a guide for decision-makers, aiming to assist in preparing patients for precise breast tumor removal while concurrently providing preoperative support [14]. We conducted a literature search on PubMed and other scientific databases using the keywords “Virtual reality”, “Augmented reality”, and “3-Dimensional Imaging” in the title field, alongside “Breast Cancer” and “Breast Operation”. Additionally, we undertook a comprehensive search on Google for information about the ‘Metaverse’, considering it as an emerging technology with limited research published on PubMed. By compiling information about the virtual versions of the physical world offered by these new technologies, we aim to provide guidelines for surgeons.

## 3. Technologies

We identified and thoroughly examined pertinent research studies, drawing conclusions from them to provide a comprehensive summary on optimizing the pre-operative diagnosis and management of breast cancer.

### 3.1. 3D Printing 

Medical 3D printing is increasingly utilized in clinical and research-based healthcare activities [15]. Three-dimensional (3D) printing involves the transformation of two-dimensional (2D) virtual data into three-dimensional objects [16]. It has revolutionized the traditional understanding of illness and treatment, particularly in the field of breast cancer [6]. By utilizing a patient’s imaging data, 3D-printed medical products can be created, facilitating the production of the physical replicas of anatomical structures [17]. 

#### 3.1.1. Materials

The materials utilized in 3D printing exhibit a range of physical and mechanical properties, with the selection dependent on the employed technique and the desired final products. These materials may exist in various forms, including liquid, paste, powder, or solid sheets, and encompass polymers, ceramics, metals, resins, and even food. Solid polymers find common usage in 3D printing techniques such as fused deposition modeling (FDM), selective laser sintering (SLS), and stereolithography (SLA). Another category of 3D printing, known as bioprinting, involves the use of polymeric hydrogels loaded with cells [18].

#### 3.1.2. Advantages and Disadvantages

The benefits of 3D printing are widely recognized and established. One application involves creating a bra-like plastic form that matches the surface of the breasts when the patient is in the supine position [19]. This form is generated through breast scanning using MRI, with the information displayed digitally for the surgeon’s reference. A 3D tracker enables precise tumor localization, allowing doctors to easily follow this image-guided process in the operating room. Advantages associated with this method include rapid prototyping, accessibility, structural control, and cost-effectiveness.

MRI-optical-guided breast-conserving surgery utilizing 3D printing offers a precise and accurate solution for removing breast tumors. This technique provides 3D spatial information of the entire tumor volume, in contrast to the single reference point provided by wire localization. A comparison study between patients undergoing wire localization and supine MRI-guided breast-conserving surgery showed that the positive margin rate and resected tissue volume were lower in the MRI-guided excision group [20].

The removed tissue from the MRI-guided group was volumetrically smaller (27.5 cm^3^) than that in the standard BCS group (57.6 cm^3^). The positive margin rate was lower in the new technique group (12.5%) than in the standard BCS group (39.3%) [20]. The gathered results, created using the data from the study by Sakakibara S. et al., are shown in Figure 1 [20].

Optical scanners allow for the adjustment of MRI scans of the breasts onto the patient on the operating room (OR) table. This approach facilitates the separation of preoperative imaging and surgery, allowing them to occur on different days [21]. The 3D printing technique, coupled with intraoperative tracking technology, ensures the localization of breast cancer with utmost accuracy [21]. A study on the 3D printing technique reveals that the distance of the predicted tumor edges in imaging is very precise, measuring <1 cm [21]. In prospective trials, lumpectomy of palpable cancers tends to be imprecise, resulting in positive margins in 28–29% of cases [22,23]. The likelihood of navigation system mismatch is very low, minimizing errors. The real-time 3D virtual reality navigation system with an open MRI is deemed feasible for safe and accurate excision of nonpalpable MRI-detected breast tumors [24]. MRI-optical-guided breast-conserving surgery based on 3D printing devices provides more precise tumor removal, offering the opportunity to avoid re-admission. This not only reduces additional stress for the patient but also cuts down on extra costs for the hospital associated with removing positive margins in the breasts. The combination of MRI with 3D mesh proves to be a convenient tool for improving the capacity of rapid prototyping and detecting invasive cancer and ductal carcinoma in situ (DCIS) [15,25,26,27].

### 3.2. Augmented Reality 

Augmented reality is an enhanced, interactive version of the real world achieved through digital visual elements, sounds, and other sensory stimuli via holographic technology. AR incorporates features such as an accurate combination of the digital and physical world and the interactions made between them in real time [28]. AR enables the superimposition of virtual reality onto clinical images of a real patient in real time. This allows visualization of internal structures through overlying tissues, providing a virtual transparency vision of surgical anatomy [29]. AR is based on hardware components including Hololens^®^ (Microsoft Corporation, Redmond, WA, USA), digital cameras/optical sensors, and laptops [30].

#### Techniques—Advantages and Disadvantages

AR superimposes 3D information onto a real environment, providing a sense of reality and immersion [31,32]. The healthcare sector demonstrates substantial savings in costs and time through the implementation of AR. AR systems heavily rely on user immersion and the interaction between the user and the system, facilitating the adoption of virtuality [33]. The research by Besl P.J. et al. [34] describes the usage of the Region of Interest (ROI), a portion of the image prepared for further measurements. The concept of ROI is employed in medical imaging to set the boundaries of breast tumors. It is based on the results detected by the vertices in the virtual coordinate system, facilitating the localization of breast tumors. This technology enables the preparation of the surgical framework, consisting of two meshes: the real coordinate system acquired through a 3D scan of the patient and the mesh on the virtual coordinate system acquired from 3D breast CT. Meshes from both the real and virtual coordinate systems are aligned initially by matching their centers of mass, and precise localization is performed using the Iterative Closest Point (ICP) technique. ICP-based registration between the patient’s 3D scan mesh and the CT-derived mesh ensures precision in breast tumor removal (Figure 2). Furthermore, since both meshes use data from the same patient, there is no need for scale adjustment, saving valuable time [34,35]. 

The ICP method [34] is among the most commonly used registration techniques due to its intuitive system and low complexity. To prepare the operating room environment, registration vertices are selected by defining the Region of Interest (ROI) through a gradient-based search method focused on the nipple area [34]. The markerless AR surgical framework introduces a new approach compared to the original process of diagnosing breast tumors. By applying this transformation, integrated visualization of medical information can be achieved by superimposing it onto the patient’s body in real space. The proposed technique allows for rapid and precise collection of AR information while minimizing patient discomfort by eliminating the need for marker placement. This study’s proposed method shows that 3D AR visualization of medical data on the patient’s body is possible using a single depth sensor without markers, achieving high accuracy with a low error mismatch. This indicates that the proposed method is more practical for real-world clinical applications compared to conventional marker-based methods [35]. Although extensive literature exists on 3D breast simulation and breast tissue modeling, only a few studies have explored the fusion of breast magnetic resonance imaging (MRI) with standard 3D scans to simulate real breast anatomy [37,38,39]. MRI is a highly precise imaging technique, and breasts reconstructed with implants can be accurately diagnosed using MRI alone. Consequently, MRI-based AR has the potential to be highly profitable. However, further investigations into this method are warranted for a comprehensive understanding and optimal utilization [35]. As an example, during surgery, an experimental digital and non-invasive intra-operative localization method with augmented reality combined with MRI was compared with the standard pre-operative localization with carbon tattooing. The augmented reality headset (Hololens^®^) visualized the precise tumor location and its projection inside the patient’s body [40]. A digital non-invasive method for intra-operative breast cancer localization using augmented reality to guide breast-conservative surgery provides local control, with free margins after tumor excision and a good cosmetic outcome [40]. The proposed MRI technique has the potential to improve the surgeon’s visualization of the tumor while enhancing the patient’s quality of life (no pain, no anxiety, no bleeding) [40]. Moreover, AR with MRI brings benefits in terms of accuracy, precision, cost, and operating time [37]. However, it should be noted that the technique based on AR with MRI is not feasible for the detection of tumors not seen on ultrasound [37]. An issue is the breast tissue deformation that occurs during MRI acquisition, for instance, due to skin contact with the scanning device, leading to inaccurate measurement [38]. Considering another approach, a fiber optic–acoustic guide (FOG) with AR for sub-millimeter tumor localization brings intuitive detection of breast tumors with minimal interference. FOG is preoperatively implanted in the tumor. When subjected to external pulsed light excitation, the FOG emits acoustic waves through its optoacoustic nano-composite layer at the tip [41]. An AR system measures the results from the ultrasound sensors and transforms the FOG tip’s position into visual feedback with <1 mm accuracy. This allows surgeons to directly visualize the tumor location and perform fast and accurate tumor removal [41]. The high rate of reoperation after breast cancer surgery, approximately 25–37%, necessitates a method that localizes the tumor and lowers the positive margin rate [42,43,44,45]. FOG under AR guidance may help breast surgeons with tumor localization and resection optimization. FOG guided by AR allows precise and fast tumor removal, significantly reducing reoperation rates and shortening surgery time [41]. In clinical practice, it is important to emphasize that sound speed differs due to tissue inhomogeneity, contributing to the localization error of breast tumors. Acoustic waves travel faster in malignant tissue, with a sound speed in malignant tissue of ~1552 mm/s and in normal tissue of ~1472 mm/s [46]. The difference in sound speed is caused by changes in breast tissue properties and may pose some detection challenges. 

Shifting to another topic, breast cancer-related lymphedema (BCRL) is a condition characterized by fluid accumulation in the upper limb of breast cancer patients who have previously undergone axillary surgery and/or radiation. Its etiology is multifactorial, including tumor-specific pathological features such as lymphovascular invasion (LVI) and extranodal extension (ENE). BCRL occurs in approximately 20–80% of breast cancer patients with lymph node metastases (N > 1) [47,48]. This condition results in swelling of the limb, leading to reduced functionality, an increased risk of comorbidities, and feelings of anxiety and depression [49,50]. Recently, augmented reality methods, such as the three-dimensional laser scanner (3DLS), have emerged as promising tools for measuring upper limb lymphedema. The 3DLS performs a surface scan of the upper limb, creating a precise 3D model. It projects a laser dot onto the upper limb, and the sensor measures the distance to the surface. This technology is cost-effective, user-friendly, reproducible, and extremely precise [51,52]. The integration of 3DLS into the clinical diagnosis of BCRL enables precise and reliable measurements. This technology is safe, painless, and can be used in various clinical settings, such as outpatient clinics or home care, potentially improving current healthcare and reducing costs in terms of personnel and procedures. The implementation of 3DLS in real-life clinical settings has the potential to significantly impact lymphedema management and patient rehabilitation. This study provides evidence that augmented reality tools, such as 3DLS, can be seamlessly integrated into the clinical assessment of BCRL, offering precise, reliable, and cost-effective diagnoses [53]. However, it is essential to consider the acquisition procedure of the 3DLS; incomplete images or errors in volumes could pose potential issues. To overcome this, the patient must hold the upper limb abducted at a 90° angle to avoid inaccuracies in detecting the limb’s volume. Moreover, the integration of AR in breast imaging offers promising advancements in detecting Occult Breast Cancer (OBC), an infrequent condition characterized by axillary metastatic carcinoma without an identifiable primary breast lesion [54,55]. In general, these new technologies extend their significance beyond diagnostic purposes alone, showcasing remarkable therapeutic potential. Notably, augmented reality guidance for intraoperative indocyanine green (ICG) lymphography during modified simplified lymphovenous anastomosis has demonstrated a promising utility as a highly effective treatment approach with minimal complications [56].

The AR technique is noteworthy for providing surgeons with opportunities to enhance their knowledge through training simulators, such as Clinical Breast Examinations (CBEs), which are physical assessments of breasts conducted by doctors for detecting tumors. A training simulator incorporating AR technology enables medical providers to assess their CBE skills. The AR application consists of two modules: one for visualizing breast deformation and the other for supporting the visualization of the palpation force applied by the user on the breast phantom [57]. CBE plays a crucial role in the early detection of breast tumors. This AR tool offers doctors valuable training, enabling them to acquire the necessary skills for precise breast tumor diagnoses.

### 3.3. Virtual Reality

Virtual reality (VR) is a computer-generated environment that simulates reality, offering users an immersive experience. This virtual world is accessed through a device called a Hololens^®^ headset. The term “VR” was first coined in 1987 by Jaron Lanier, a researcher and engineer who made substantial contributions to the VR industry [58]. VR creates an immersive experience that allows users to detach from the real world [28]. Unlike augmented reality (AR), VR completely immerses users in a virtual environment, opening up new possibilities.

#### Techniques—Advantages and Disadvantages

A multidimensional virtual system was developed to simultaneously display morphological and functional tissue information [56]. Spatial representation of the computed breast tissue offers several possibilities: improved contrast enhancement; inspection of the data volume in 3D space with the ability to rotate, apply transparency, and localize lesions in space, enabling fast topological recognition. VR is a method for fast and efficient analysis of dynamic MR imaging of female breasts. Specifically, it holds significant potential in the detection and localization of multiple breast lesions [59]. Thanks to VR and magnetic resonance (MR) imaging, morphologic and functional tissue information on the breasts can be displayed simultaneously. The combination of VR and MR imaging not only reproduces reality but also intensifies the visual representation of the data. VR, in a version of the computer software “3D Slicer (version 5.0),” can generate a 3D virtual reality model of the tumor, providing guidelines for precise needle puncture in the breast [24]. “3D Slicer” is software for visualization, processing, and segmentation. Surgeons can use this system to enhance breast analysis, leading to improved surgical outcomes. Surgeries can be performed with better accuracy, and 3D Slicer facilitates this improvement [60]. MRI was confirmed to be clinically important in the diagnosis of DCIS [61]. Breast MRI can be an ideal modality for image-guided surgery for DCIS; hence, surgical treatment can be combined with this technique in the near future [62]. Therefore, using a three-dimensional (3D) virtual reality navigation system with MRI for breast-conserving surgery gives the possibility to remove the tumor with the highest precision and minimal error [24]. This system requires expertise in breast surgery and training in navigating a 3D virtual environment. It aids surgeons by enhancing tumor localization and removal.

Metastatic Breast Cancer (MBC) poses a challenge for doctors to provide precise treatment while maintaining the patient’s quality of life [63]. People with MBC commonly report adverse events from cancer treatments [64]. Symptoms are both physical (pain, fatigue [65]) and psychological (depression, anxiety [66]). MBC significantly impacts the quality of life and can be a source of suffering [65]. Common therapies to treat the symptoms include rehabilitation programs [67] and psychosocial interventions [68]. However, a more precise approach is required. Women with MBC often report physical and psychological symptoms that affect their quality of life. Immersive VR has been proposed as an adjunctive pain therapy for cancer patients [69]. VR presents an interactive 3D human–computer interface that allows individuals to interact with it and become immersed in a computer-generated environment naturally [70]. An example of an immersive VR application is ‘Snow World’, in which patients take a virtual journey down an icy river [71]. This application has been shown to effectively reduce pain in patients who undergo burn wound debridement [71]. Consequently, immersive VR has been proposed as an adjunctive therapy for reducing cancer-related symptoms [72]. The impact of VR on chronic pain is very significant. VR interventions improve a patient’s quality of life, reducing fatigue, pain, depression, anxiety, and stress. Fatigue in cancer patients is associated with a higher level of pro-inflammatory cytokines [73] and chronic inflammation [74], which promote tumor growth [75]. VR interventions may potentially improve immune function and response to the treatment [75,76,77]. Consequently, controlling fatigue and stress levels can influence tumor control. The VR session is short, allowing the patient to immerse themselves in pleasant virtual scenes. VR interventions effectively reduce patient anxiety, thereby impacting their fatigue and pain levels. This improvement in well-being also boosts their immune system. An additional benefit is that VR interventions can be conducted in a patient’s own home. Previous VR studies were conducted in hospitals, where the surrounding environment often exacerbated anxiety and other symptoms [78]. Conducting VR interventions at home is more convenient for patients. According to research written by Reynolds L. et al. [69], participants who completed a two-week-long VR experience immediately reported symptoms. By analyzing the collected results, it is evident that VR has a visible impact on pain control. These beneficial effects have been demonstrated in different clinical applications including chemotherapy [79], painful procedures [80], and hospitalization [81]. VR should be considered a viable and acceptable adjunctive therapy to alleviate physical and psychological symptoms in cancer patients [69]. Upon analyzing the data, it is evident that VR interventions significantly impacted the patients’ lives. The Standardized Mean Difference value provides evidence that VR has a positive influence on patients [82] (Figure 3).

### 3.4. Metaverse

The concept of the virtual reality metaverse was introduced by Neal Stephenson in his 1992 novel titled “Snow Crash”, combining the words ‘meta’ and ‘universe’ [83]. The term “metaverse” combines “meta”, which means “virtual and transcendent”, and “universe”, denoting the world [84]. The metaverse has been widely discussed following Facebook’s announcement of changing its company name to ‘Meta’. This artificial world is broadly defined as a virtual technology accessed by individuals via AR, VR, mixed reality, and extended reality [85]. The metaverse is a digital world established on the integration of virtual and real-world technologies. Individuals can enter the metaverse using digital identities [86]. This concept can be understood by analyzing other virtual spaces which use avatars such as Roblox [87] and Fortnite [88]. The metaverse has numerous potential benefits in cancer care. It provides immersive VR therapy for cognitive assessment, rehabilitation, and digital evaluation [89]. Overall, the metaverse is considered to be the future healthcare platform [90]. Research by Koo H. et al. [91] displays the beneficial effect of the metaverse on the healthcare system. It is said that the 2019 coronavirus pandemic has made it challenging to conduct medical training. Observing high-tech medical equipment and surgeries for learning purposes has become nearly impossible. In response, educational methods involving the metaverse have been introduced in the medical field, providing non-face-to-face education for medical staff worldwide. Participants in the program wore a headset to experience a real place in a virtual environment. Surgeries were broadcast and participants could observe the procedure in an actual operating room with surgeons, surgical nurses, and the 360° environment through the headset. The learning environment was more accurate than from direct observation in the operating room [92]. Moreover, the number of people who can physically enter an operating room is limited, while in the metaverse, there are no limits. This demonstrates the positive impact of the metaverse on the healthcare system. Through this technology, observers can experience the process in a more extensive and precise manner. Chand, M. highlights that gaining practical knowledge is a must for surgeons who are learning their craft. Interacting in a 3D environment provides the opportunity to enter the operating theater in a more prepared state, having been ‘pre-trained’ through VR. To have an immersive experience, it is crucial to provide the same view from the operating room via the VR headset. The metaverse facilitates training with easy access to high-quality education and surgeries worldwide. Additionally, the technology enables online consultations between doctors and patients [93]. The metaverse was presented as a revolutionary virtual world, introducing possibilities such as the ability to collaborate on a project seamlessly across different countries. Metaverse Microsoft Mesh can save all information in “cloud” storage with easy access to data banks. The immersive 3D world with AR and VR can be used for business, marketing, and advertising new medical products. The metaverse can also provide support for mental and physical health [94]. It is important to emphasize that the metaverse is still under development, and to keep data safe, proper security is needed. MetaBreast is a metaverse specifically correlated to breast cancer surgery. Pedro Gouveia once said that MetaBreast is an evolving project for planning breast cancer surgeries. AR and VR together provide visualization of the breast tumor and its margins during surgery to localize the tumor with precision. This increases surgical accuracy, allowing for the complete extraction of the breast tumor during a single surgical operation. According to Ana Gerschenfeld, Health and Science Writer of the Champalimaud Foundation, the metaverse has the potential to improve the procedure of breast tumor removal. MetaBreast develops a new digital diagnostic tool to superimpose the radiological data onto the patient’s body in the operating room. MetaBreast consists of the software, operative room, and the collaboration between trained doctors and engineers [95]. BodySCULPT^®^ (New York City, NY, USA) is one of the plastic surgery practices in New York City which is based on the metaverse world. The clinic has accredited practice for the use of the Oculus Rift 3D imaging glasses for consultations. Dr. Spero Theodorou, the plastic surgeon at bodySCULPT, mentioned that the relationship between the plastic surgeon and the patient is now enhanced to the point of bringing the virtual as close to reality as technically possible [96]. Oculus Rift 3D^®^ (Menlo Park, CA, USA) and Hololens^®^ provide the immersive visualization of complex anatomy based on a CT scan. Real-time tracking ensures the accurate fusion of the scan with the patient’s body regardless of any changes in position. For precise visualization, it is important to measure the pupilar distance from the eye to the Hololens^®^ glasses. The installation of additional sensors in the glasses can allow for the measurement of the doctor’s eye position and view. The minimization of accommodation errors provides more accurate depth of visualization. The increased field of view of the video camera allows the observer to see the holographic content from more angles. The higher-resolution camera can provide more accurate patient tracking. A change in the patient’s weight has no influence on the measured error [97]. In the research written by Gary Masterton, a new technique under the name of the Computer Tomography Angiography (CTA) was discussed. HoloLens permits surgical planning to identify the patient’s arteries and perforators. This technology allows the surgeon to superimpose the CTA images directly onto the patient, facilitating an “in vivo” picture of the underlying anatomy prior to the incision. This technique provides the support for surgeons to identify patients’ vasculature to facilitate accurate dissection, optimize surgical outcomes, and minimize morbidity. Real-time surgical planning enhances the accuracy and efficiency of the operative technique by using the pre-operative CTA imaging. This technology demonstrates the potential to revolutionize the future practice of free flaps for breast reconstruction [98]. In conclusion, the metaverse is still a new and evolving technology. Further exploration and research are necessary to fully understand its potential. It is important to emphasize that this technology offers numerous advantages to the healthcare system.

## 4. Conclusions

The comprehensive analysis presented in this research underscores the transformative potential of cutting-edge technologies in reshaping the landscape of breast cancer diagnosis and management. The integration of innovative tools, including 3D imaging, augmented reality (AR), virtual reality (VR), and the metaverse, holds the key to achieving a remarkable precision rate with minimal errors during breast cancer operations. These technologies have proven instrumental in not only enhancing the detection of breast cancer but also significantly reducing the false negative rate and elevating overall diagnostic precision. Of particular significance is the profound impact on lymphatic complications associated with breast cancer treatment. Digital innovations offer a unique way to address and mitigate these challenges, showcasing the potential to revolutionize not only the diagnostic process but also the therapeutic strategies employed. The continuous improvement in breast tumor detection techniques necessitates the seamless integration of these technologies, aligning with the latest research findings and guidelines. In conclusion, the synergy of a multidisciplinary approach and groundbreaking digital innovations stands poised to revolutionize breast cancer diagnosis and therapy.

## Figures and Tables

**Figure 1 jcm-13-00915-f001:**
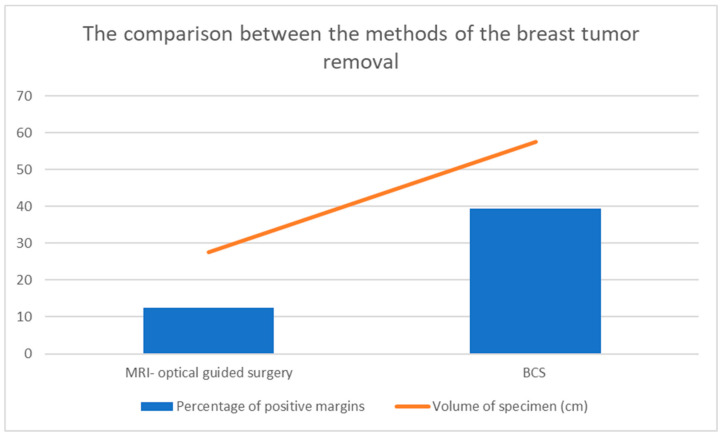
Differences between MRI-optical-guided surgery and classical BCS method [20].

**Figure 2 jcm-13-00915-f002:**
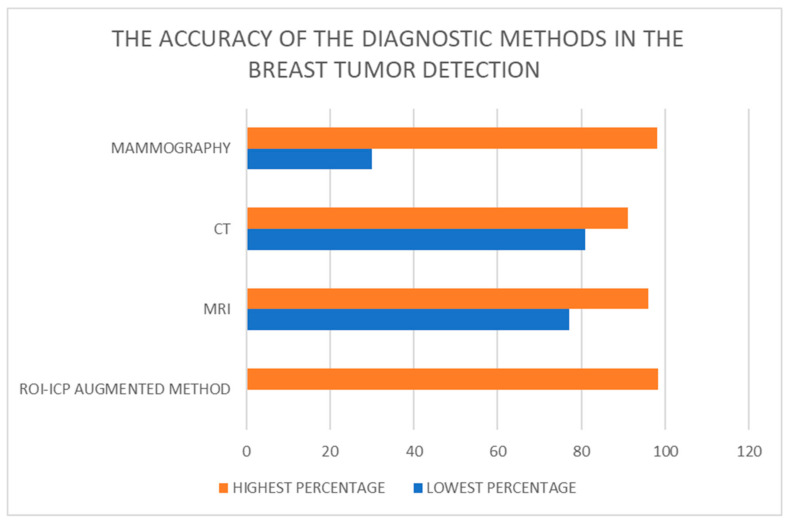
Comparison of breast tumor diagnosis accuracy across different methods [25,34,36].

**Figure 3 jcm-13-00915-f003:**
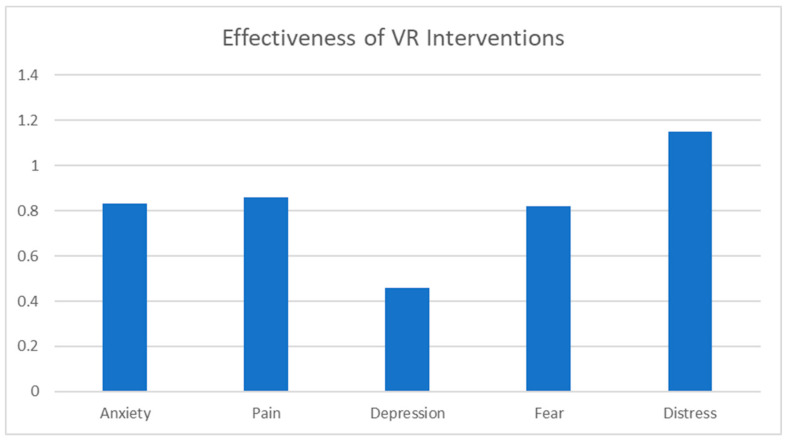
The effects of VR-based interventions on the mental health of the patients, expressed by Standardized Mean Difference (SMD).

## Data Availability

Not applicable.

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
