# Peer review of "Current Trends and Beyond Conventional Approaches: Advancements in Breast Cancer Surgery through Three-Dimensional Imaging, Virtual Reality, Augmented Reality, and the Emerging Metaverse"

_jcm, 2024, doi:10.3390/jcm13030915_

Round 1

Reviewer 1 Report

Comments and Suggestions for Authors

The manuscript presents a comprehensive and insightful exploration into the integration of advanced technologies like 3-Dimensional Imaging, Virtual Reality (VR), Augmented Reality (AR), and the Metaverse in breast cancer surgery. It offers a forward-looking perspective on how these technologies could revolutionize surgical procedures, diagnosis, and patient management in the field of breast cancer, a subject of great importance and relevance in contemporary oncological practice.

One of the manuscript's strengths lies in its detailed examination of each technology. The sections on 3D printing, AR, VR, and the Metaverse are thoroughly researched and articulate the current state and potential future applications of these technologies in breast cancer treatment. The discussion on how these technologies can enhance surgical precision, improve patient outcomes, and potentially reduce the need for re-operation is particularly noteworthy. Additionally, the manuscript successfully illustrates the interdisciplinary nature of modern surgical practice by intertwining technological advancements with clinical applications.

However, there are areas in the manuscript that warrant improvement. The introduction, while informative, appears overly generalized and lengthy. A more concise and focused introduction, emphasizing the specific advancements and their direct implications in breast cancer surgery, could enhance the overall impact and clarity of the paper.

Moreover, it would be highly beneficial for the authors to extend their investigation into the impact and support of innovative technologies, such as Virtual Reality, Augmented Reality, and 3-Dimensional Imaging, in the challenging areas of breast cancer treatment pertaining to Cancer of Unknown Primary (CUP) Syndrome and metastatic breast cancer ab-initio. These specific areas represent critical aspects of breast cancer management where the application of advanced technologies could offer significant improvements in diagnosis and treatment strategies. The inclusion of insights and findings from pivotal studies in these fields, particularly those reported in PMID: 33173479 and PMID: 36551722, would not only enrich the manuscript but also provide a more comprehensive understanding of the potential of these technologies in addressing complex and advanced stages of breast cancer. This additional focus would underscore the manuscript's relevance to a broader spectrum of clinical scenarios in breast cancer care, thereby enhancing its contribution to the field.

Additionally, the manuscript would benefit from moderate revision for language and clarity. While the technical content is strong, the flow and readability of the text are occasionally hindered by minor grammatical inconsistencies and awkward phrasings.

Comments on the Quality of English Language

The manuscript would benefit from moderate revision for language and clarity. While the technical content is strong, the flow and readability of the text are occasionally hindered by minor grammatical inconsistencies and awkward phrasings.

Reviewer 2 Report

Comments and Suggestions for Authors

Review to: „Current trends and beyond conventional approaches: advancements in breast cancer surgery through 3-Dimensional Imaging, Virtual Reality, Augmented Reality, and the emerging Metaverse“

Overall impression:

The manuscript discusses novel digital techniques in breast cancer surgery.

Strengths:

1.       Clear Objective: The manuscript clearly outlines the objective of introducing new digital tools and provides a comprehensive description of innovative methods.

2.       Detailed technical procedures: Technical procedures are thoroughly explained, supported by figures, which enhances the clarity oft he manuscript.

3.       This manuscript has a very interesting topic, as it proposes an overview of actual innovations in the field of digital imaging.

Areas of improvement:

1.       Section Material and Methods: In line 57 „Imagine“ should be „imaging“

2.       Section 3.2.2 Augmented reality helps not only in diagnosis of lymphoedema but even in the surgical treatment (Brebant V, Heine N, Lamby P, Heidekrueger PI, Forte AJ, Prantl L, Aung T. Augmented reality of indocyanine green fluorescence in simplified lymphovenous anastomosis in lymphatic surgery. Clin Hemorheol Microcirc. 2019;73(1):125-133. doi: 10.3233/CH-199220. PMID: 31561348.)

3.       Section Conclusion: The conclusion could be strengthened by emphasizing the potential impact of the digital innovation in the diagnosis and therapy of breast cancer and could help avoiding and treating lymphatic complications of breast cancer treatment.

This review consolidates impactful research findings, offering a holistic perspective on the role of advanced imaging technologies in breast cancer management. By presenting a comprehensive analysis of Virtual Reality, Augmented Reality, 3-Dimensional Imaging, and the emerging metaverse, the article contributes valuable insights for healthcare professionals striving to enhance the diagnosis and treatment of breast cancer. My suggestions aim to enhance the manuscript’s clarity, completeness and impact.
